# The Novel Oral mTORC1/2 Inhibitor TAK-228 Reverses Trastuzumab Resistance in HER2-Positive Breast Cancer Models

**DOI:** 10.3390/cancers13112778

**Published:** 2021-06-03

**Authors:** Marta Sanz-Álvarez, Ester Martín-Aparicio, Melani Luque, Sandra Zazo, Javier Martínez-Useros, Pilar Eroles, Ana Rovira, Joan Albanell, Juan Madoz-Gúrpide, Federico Rojo

**Affiliations:** 1Department of Pathology, Fundación Jiménez Díaz University Hospital Health Research Institute (IIS—FJD, UAM)—CIBERONC, 28040 Madrid, Spain; marta.sanza@quironsalud.es (M.S.-Á.); ester.martin@fjd.es (E.M.-A.); melani.luque@quironsalud.es (M.L.); szazo@fjd.es (S.Z.); 2Translational Oncology Division, OncoHealth Institute, Health Research Institute-Fundación Jiménez Díaz (IIS-FJD, UAM), 28040 Madrid, Spain; javier.museros@quironsalud.es; 3Institute of Health Research INCLIVA-CIBERONC, 46010 Valencia, Spain; Pilar.Eroles@uv.es; 4Department of Physiology, University of Valencia, 46010 Valencia, Spain; 5Cancer Research Program, IMIM (Hospital del Mar Research Institute), 08003 Barcelona, Spain; arovira@imim.es (A.R.); 96087@parcdesalutmar.cat (J.A.); 6Medical Oncology Department, Hospital del Mar-CIBERONC, 08003 Barcelona, Spain; 7Department of Experimental and Health Sciences, Faculty of Medicine, Universitat Pompeu Fabra, 08002 Barcelona, Spain

**Keywords:** breast cancer, resistance, anti-receptor therapy, trastuzumab, PI3K, mTOR, TAK-228

## Abstract

**Simple Summary:**

Hyperactivation of the PI3K/AKT/mTOR cell signalling pathway is an important and well-described mechanism of trastuzumab resistance in HER2-positive breast cancer. In cell-line models of acquired trastuzumab resistance generated in our laboratory, we demonstrate this type of activation, which is independent of HER2-mediated regulation. We investigate whether the use of specific mTOR inhibitors, a PI3K/AKT/mTOR pathway effector, could lead to decreased activity of the pathway, influencing trastuzumab resistance. We demonstrate that TAK-228, a mTORC1 and mTORC2 inhibitor, can reverse resistance and increasing response to trastuzumab in models of primary and acquired resistance.

**Abstract:**

The use of anti-HER2 therapies has significantly improved clinical outcome in patients with HER2-positive breast cancer, yet a substantial proportion of patients acquire resistance after a period of treatment. The PI3K/AKT/mTOR pathway is a good target for drug development, due to its involvement in HER2-mediated signalling and in the emergence of resistance to anti-HER2 therapies, such as trastuzumab. This study evaluates the activity of three different PI3K/AKT/mTOR inhibitors, i.e., BEZ235, everolimus and TAK-228 in vitro, in a panel of HER2-positive breast cancer cell lines with primary and acquired resistance to trastuzumab. We assess the antiproliferative effect and PI3K/AKT/mTOR inhibitory capability of BEZ235, everolimus and TAK-228 alone, and in combination with trastuzumab. Dual blockade with trastuzumab and TAK-228 was superior in reversing the acquired resistance in all the cell lines. Subsequently, we analyse the effects of TAK-228 in combination with trastuzumab on the cell cycle and found a significant increase in G0/G1 arrest in most cell lines. Likewise, the combination of both drugs induced a significant increase in apoptosis. Collectively, these experiments support the combination of trastuzumab with PI3K/AKT/mTOR inhibitors as a potential strategy for inhibiting the proliferation of HER2-positive breast cancer cell lines that show resistance to trastuzumab.

## 1. Introduction

Despite ongoing advances in understanding diagnosis and treatment, breast cancer continues to place an enormous burden on healthcare systems worldwide and poses a risk to the lives of many patients. Breast cancer is the second leading cause of cancer deaths among women worldwide, representing 30% of all new cancer diagnoses: More than 2.25 million new cases and around 700,000 deaths were estimated in 2020 [1]. Breast cancer is a heterogeneous disease comprising four major subtypes, each with distinct pathological features and clinical implications [2]. Among those subgroups, HER2-positive breast cancer accounts for 25% of all cases and is associated with high relapse rates and poor prognosis [3,4]. This subtype is characterised by amplifying the *ERBB2/neu* oncogene and/or overexpression of its associated HER2 tyrosine kinase receptor [5]. Despite the absence of a ligand for this transmembrane receptor, HER2 forms homodimers or heterodimers with other HER family members, activating different downstream signalling pathways, including MAPK and PI3K/AKT/mTOR, which ultimately regulate processes, such as cell survival, proliferation, motility and metabolism [6,7]. In 1998, the advent of trastuzumab, the first targeted anti-HER2 therapy and humanised monoclonal antibody against HER2, brought about considerable improvement in the prognosis of metastatic and early-stage HER2-positive breast cancer patients [8,9]. In spite of the efficacy demonstrated by trastuzumab, both alone and in combination with chemotherapy as first-line treatment, primary or acquired resistance emerges within a few months after the start of treatment, and resistance remains one of the main problems in managing these patients [8,10]. Several mechanisms of resistance to trastuzumab have been described in recent decades, such as the expression of splicing variants like p95HER2 [11], heterodimerisation with other RTKs [12,13,14], Src activation [15] and aberrant activation of the PI3K signalling pathway, most commonly through mutations in PIK3CA and loss of PTEN [16,17]. The intertwining of HER2-mediated signalling and the PI3K pathway takes the form, at the molecular level, that signalling by the HER family is primarily mediated through the PI3K and MAPK cascades [18,19]. As a result, the PI3K/AKT/mTOR signalling pathway has been implicated in the anti-HER2 response [17,20,21], and targeting the PI3K/AKT/mTOR pathway has proven to be a valuable strategy to overcome resistance to HER2-directed therapy [22].

Due to the involvement of the PI3K pathway in both HER2-mediated signalling and in the emergence of resistance to trastuzumab, this network becomes a good target for drug development. Because inhibition of the PI3K/AKT/mTOR axis results in enhanced HER2 signalling in HER2-overexpressing breast cancer, especially increased expression of HER2 and HER3 [23], targeting both pathways could prevent the development of resistance. However, the clonal evolution of cancer itself causes genetic and molecular diversity in patients’ tumours that manifests as long-recognised functional and phenotypic heterogeneity. It is, therefore, unclear whether, in a HER2-positive breast cancer subtype scheme, such a therapeutic combination will be effective in different scenarios characterised by small molecular variations, this despite previously published reports in the scientific literature. As reported elsewhere [24], our laboratory generated and characterised several cellular models of trastuzumab-resistant HER2-positive breast cancer lines, covering, albeit to a limited extent, a range of genetic heterogeneity. Moreover, several drugs that are effective against different nodes of the PI3K/AKT/mTOR signalling pathway are available, namely, BEZ235, everolimus, and TAK-228. Different preclinical studies have demonstrated the efficacy of combining trastuzumab with different PI3K/AKT/mTOR inhibitors. For instance, BEZ235, a dual pan-class I PI3K and mTOR kinase inhibitor, has shown antitumor activity in vitro and in vivo in breast cancer models that harbour PI3KCA mutations [25] or are resistant to anti-HER2 therapies [26]. In murine models of HER2-positive mammary tumours, combined therapy with trastuzumab and everolimus, an allosteric mTORC1 inhibitor, obtained better results than either agent alone [27]. Furthermore, in a resistance model generated by the loss of PTEN, trastuzumab combined with everolimus restored sensitivity to trastuzumab and showed greater efficacy than either agent independently [28]. TAK-228 is an ATP-competitive inhibitor that targets both mTORC1 and mTORC2. TAK-228 has shown efficacy in different preclinical models of breast cancer [29,30]. The aim of our study was to evaluate the efficacy of three different mTOR inhibitors in in vitro models of trastuzumab-resistant breast cancer cells to assess their potential use in both primary resistance and the development of acquired resistance. We show that trastuzumab, in combination with mTOR inhibitors, exerts an antiproliferative effect by inducing alterations in the PI3K/AKT/mTOR and ERK pathways, as well as through the induction of apoptosis and cell cycle arrest in different models of trastuzumab resistance. Our data suggest a potential benefit of using mTOR inhibitors in combination with trastuzumab in acquired resistance.

## 2. Materials and Methods

### 2.1. Cell Lines

The effects of trastuzumab on cell growth were studied in a panel of eleven HER2-amplified breast cancer cell lines, including four trastuzumab-conditioned cell lines selected for long-term outgrowth in trastuzumab-containing medium. BT-474 (HTB-20) ductal carcinoma, SK-BR-3 (HTB-30) and AU-565 (CRL-2351) adenocarcinoma, as well as HCC1419 (CRL-2326) and HCC1954 (CRL-2338) ductal carcinoma cell lines, were obtained from the American Type Culture Collection. EFM-192A (ACC-258) and JIMT-1 (ACC-589) ductal carcinoma cells were obtained from the German Tissue Repository DSMZ. Trastuzumab-resistant BT-474.rT3, SK-BR-3.rT1, AU-565.rT2 and EFM-192A.rT1 cell lines were generated as previously described [24]. BT-474, SK-BR-3 and JIMT-1 cells were maintained in DMEM-F12 supplemented with 10% heat-inactivated foetal bovine serum (FBS), 2 mmol/L glutamine, and 1% penicillin G-streptomycin. AU-565, HCC1419 and HCC1954 cells were cultured in RPMI 1640 supplemented with 10% heat-inactivated FBS, 2 mmol/L glutamine, and 1% PSF. EFM-192A cells were grown in RPMI 1640 medium supplemented with 20% heat-inactivated FBS, 2 mmol/L glutamine, and 1% PSF. Cells were maintained at 37 °C with 5% CO_2_. All cell lines were checked for authentication every 6 months, either by using the Cell Line Authentication service at LGC Standards (UK) (tracking no: 710259498; 710274855; 710281607; 710272355), or by running a home-made mutational profiling assay.

### 2.2. Reagents

The recombinant humanised monoclonal HER2 antibody trastuzumab (a concentration of 15 µg/mL was selected as indicated elsewhere [24]) (Herceptin, Genentech, San Francisco, CA, United States) was supplied by the pharmacy of our hospital; BEZ235 (S1009), everolimus (S1120) and TAK-228 (S2811) were obtained from Selleckchem (Selleckchem Spain, Madrid, Spain).

### 2.3. Determination of the Resistance Rate

Establishment of drug resistance was confirmed by cell proliferation assay, as determined in P100 plates containing 5 × 10^5^ cells for each condition (sensitive and resistant), grown both in the absence and in the presence of trastuzumab for 7 days. The results were processed using the algorithm described by O’Brien, which correlates the rate of growth between the treated and nontreated cells, reflecting the doubling time of the cells [31]. Once resistance was confirmed, cells were maintained in the absence of treatment for 30 days. After this pause, resistance was reconfirmed using the same protocol. Resistant cell lines populations were maintained with 15 µg/mL of trastuzumab in the medium for months. Periodically, vials of both the sensitive (parental) and resistant cell populations (pools and clones) were stored in liquid nitrogen to keep a stock of young cells.

### 2.4. Cell Proliferation Assays

Cells were seeded in triplicate in P100 plates at a density of 5 × 10^5^ cells per plate and allowed to adhere and enter the growth phase before being treated with or without 15 μg/mL trastuzumab for 7 days in the appropriate culture medium. Cells were then harvested by trypsinisation and counted with trypan blue using the TC20 Automated Cell Counter (BioRad, Hercules, CA, USA). The appropriate culture media and trastuzumab were replaced every 3 days. All experiments were repeated three times with readings at least in triplicate for each concentration.

### 2.5. Determination of IC50

To determine the IC50 of the mTOR inhibitors, a panel of HER2-positive breast cancer cell lines was treated with escalating concentrations of BEZ235, everolimus and TAK-228. Proliferation was measured by counting after 7 days of treatment. Viable cells were counted by trypan blue exclusion. IC50 (half the maximal inhibitory concentration) was calculated using SigmaPlot software. Values are mean IC50 from three independent experiments.

### 2.6. Protein Extraction and Quantification

Cells were washed with 3 mL PBS at RT. Next, cells were scraped in the presence of 150 µL lysis buffer (RIPA, peptidase inhibitor, phosphatase inhibitor) at 4 °C and transferred to a 1.5-mL tube. Cells were incubated in lysis buffer for 20 min at 4 °C and sonicated afterwards. Then the cell lysate was spun at 13,000× *g* for 10 min at 4 °C, and the supernatant was retained and stored. Protein extracts were quantified using the Pierce BCA protein assay kit (Thermo Fisher Scientific, Whaltman, MA, USA), following the manufacturer’s instructions.

### 2.7. Western Blotting (WB)

Protein aliquots were prepared at 1 µg/µL in 4× Laemmli loading buffer and boiled at 95 °C for 6 min. Twenty µL of protein extract was loaded in a 10% polyacrylamide gel (SDS-PAGE). Next, proteins were transferred to a nitrocellulose membrane for 1 h at 100 V and 4 °C. The membrane was blocked (5% milk in TBST 1×) for 1 h, washed 3 times for 10 min and then incubated with the primary antibody at RT overnight under agitation. The concentrations used were as follows: HER3 (1:500; Thermo Scientific), p-HER3 Tyr1197 (1:1000), HER2 (1:500), p-HER2 Tyr1221/1222 (1:1000), AKT (1:1000), p-AKT Thr308 (1:300), p-AKT Ser473 (1:500), p44/42 MAPK (ERK1/2) (1:1000), p-p44/42 MAPK (ERK1/2) Thr202/Tyr204 (1:1000), 4E-BP1 (1:500); p-4E-BP1 Thr37/46 (1:500); p-4E-BP1 Thr70 (1:500); S6 ribosomal protein (S6) (1:500); p-S6 ribosomal protein (p-S6) Ser235/236 (1:1000) (Cell Signaling, Danvers, MA, USA) and GAPDH (1:5000; Sigma-Aldrich, St. Louis, MO, USA). All primary antibodies were rabbit, except the anti-HER3, which was mouse; all were monoclonal. Then the membranes were washed 3 × 10 min in TBST and incubated with a secondary antibody (diluted in 2.5% BSA in TBS 1×) at RT for 1 h. ECL-anti-mouse and ECL-anti-rabbit secondary antibodies attached to peroxidase (HRP; GE Healthcare, Chicago, IL, USA) were used at a concentration of 1:5000. The membranes were washed 3 × 10 min again, and immeserd in the detection reagent (ECL or ECL Prime, if applicable; Amersham, GE Healthcare) for 1 min, prior to developing on a photographic film. Densitometry and quantification of proteins were carried out using ImageJ software.

### 2.8. Flow Cytometric Determination of Cell Cycle Arrest and Apoptosis

Before carrying out cell cycle detection and apoptosis, cell lines were synchronised by serum starvation for 24 h. Cell cycle and apoptosis were analysed after treatment with either vehicle (i.e., trastuzumab 15 µg/mL, TAK-228 0.5 µM or both) for 24 and 72 h, respectively. For cell cycle arrest analysis, cells were collected after treatment, washed with PBS and fixed with 70% cold ethanol at –20 °C for at least 2 h. Cells were incubated with 0.5 mg/mL RNase (Sigma-Aldrich) at 37 °C for 30 min, and finally stained with propidium iodide (BD Biosciences, Franklin Lakes, NJ, USA) for 10 min. Apoptosis was assessed with the Annexin-V-FITC Apoptosis Detection Kit (BD Biosciences) according to the manufacturer’s instructions. Flow cytometry was performed on a FACS Canto II (BD Biosciences), and data were analysed with FACS Diva software (BD Biosciences).

### 2.9. Statistical Analysis

All data are expressed as means ± standard deviations for at least three replicates (unless otherwise indicated). Statistical significance was analysed by a two-tailed Student’s *t*-test (*: *p* < 0.05, **: *p* < 0.01, ***: *p* < 0.001). This work was performed in accordance with the Reporting Recommendations for Tumour Marker Prognostic Studies (REMARK) guidelines [32].

## 3. Results

### 3.1. Development and Characterisation of a Panel of Breast Cancer Cell-Line Models of Acquired Trastuzumab Resistance

To test the efficacy of a combination of HER2 blockade with mTOR inhibition as a potential therapeutic strategy to overcome resistance to trastuzumab in HER2-positive breast cancer cell line (BCCL) models, we first developed four different cellular models with acquired resistance to trastuzumab [24]. Briefly, we used prolonged exposure to moderate doses of the drug to generate novel BCCLs with acquired resistance to trastuzumab, authenticated them based on their molecular profile and their resistance rate was determined. We selected clones for each of the BCCLs and screened them for trastuzumab sensitivity after seven days of treatment (Figure 1). We observed that in all cases, resistant cells showed a higher growth rate in the presence of the drug than the parental sensitive cells. The biochemical analysis of the status of kinase receptors and effectors from different cellular pathways actionable by HER2 signalling revealed differences in phosphorylation levels for several targets between sensitive and resistant lines (Appendix A), as we reported previously [24]. After treatment with trastuzumab, changes occurred in the phosphorylation levels of HER2, AKT (Thr308 and Ser473), ERK1/2, and S6, with more relevant changes between sensitive and resistant populations in the BT474 and AU565 cell lines. This finding was consistent with patterns of molecular alterations commonly described in breast cancer [25]. De novo trastuzumab-resistant cell lines HCC1419, HCC1954 and JIMT-1 were also examined for biochemical changes in the HER2 and PI3K/AKT/mTOR pathways (Appendix A). The most notable signal was the abundant expression of 4E-BP1 in both cell lines, which does not appear to translate into strong activation in either case. Phosphorylation levels of S6 were not elevated either. On the other hand, we observed a slight decrease in AKT phosphorylation levels in the JIMT-1 cell line compared to HCC1954. Overall, the two lines do not show phosphorylation activation signals for either of the two pathways studied.

### 3.2. Effect of Anti-HER2 and MTORC1/2 Treatments on HER2-Positive Breast Cancer Cell Lines (Determination of IC50)

To determine the effects of BEZ235, everolimus and TAK-228 on the inhibition of the PI3K/AKT/mTOR proliferation axis in HER2-positive cells, the panel of eleven cell lines with varying sensitivity to trastuzumab was treated with increasing inhibitor concentrations. After seven days of treatment, cellular proliferation was measured to determine the IC50 for each drug and cell line (Appendix A). In general, a similar sensitivity was observed in all cell lines for every drug, so when treated with any of the three mTOR inhibitors, the proliferation of the eleven cell lines was significantly inhibited at low nanomolar ranges. The determination of sensitivity to BEZ235 showed that all cell lines behaved very similarly when exposed to the treatment, and only the SK-BR-3.rT1 line was more sensitive to this drug than its parental line. The everolimus sensitivity study showed that all the lines were sensitive to treatment at high concentrations. In addition, JIMT-1 was very sensitive to this drug, decreasing its cell proliferation by more than 50% at 1 nM everolimus, and AU-565.rT2 was also found to be more sensitive to treatment than its sensitive parental line. Finally, treatment with TAK-228 showed highly similar sensitivity to treatment in all lines, both trastuzumab-sensitive and trastuzumab-resistant. Based on these results, the IC50 was calculated for each of the lines and for each drug (Table 1). Notably, the IC50 value of everolimus was more heterogeneous between cell lines than the IC50 values of the other two drugs. In addition, the IC50 values of BEZ235 and TAK-228 between the resistant lines and their parents were very similar, though this was not the case for everolimus in AU-565.rT2 and EFM-192A.rT1, which had a significantly lower IC50 value than their respective parental cell lines. The exceptions were the effect of everolimus in HCC1419 and particularly in JIMT-1, which showed at least a 10× increased sensitivity with respect to the other cells. This is probably because different mutations in nodes of the PI3K/AKT/mTOR pathway make some cell lines more sensitive to everolimus than others, which turn out to be more resistant [33]. 

In view of these results, we considered that combining anti-HER2 therapy with each of these mTOR inhibitors might show a greater antiproliferative effect. For therapeutic studies, the concentration and time of treatments were based on previous reports, and administered as follows: Trastuzumab (15 µg/mL) [15]; BEZ235 (1 nM, 5 nM and 20 nM) [34], everolimus (0.5 nM and 1 nM) [35] and TAK-228 (1 nM and 5 nM) [30].

### 3.3. Combined Treatment of Trastuzumab and MTORC1/C2 Inhibitor TAK-228 in HER2-Positive Breast Cancer Cell Lines with Acquired Resistance to Trastuzumab

In order to assess the potential synergistic effects of trastuzumab in combination with mTOR inhibitors, we performed viability assays in the four sensitive cell lines, as well as their correspondent resistant models. Overall, the combination of trastuzumab with BEZ235 or everolimus influenced the therapeutic response to a lesser degree than the combination treatment of trastuzumab with TAK-228 because, although it causes a reduction of mTOR activation in the cell lines, cell viability was not affected. In contrast, the combination of trastuzumab with TAK-228 significantly increased the therapeutic response in all cases, suggesting that the decreased mTOR activation status by TAK-228 affects sensitivity to trastuzumab.

The treatment effect of the TAK-228 inhibitor was evaluated using two treatment concentrations (1 nM and 5 nM), as monotherapy and in combination with trastuzumab (Figure 2). A single treatment with TAK-228 showed no effect on cell proliferation in any of the cell lines for either of the two concentrations used. Combination treatment with trastuzumab and TAK-228 5 nM resulted in the reversal of acquired resistance in all lines. BT-474.rT3 cells showed a highly significant decrease in proliferation in the trastuzumab and TAK-228 condition (52%) compared to trastuzumab (84%, *p*-value < 0.01) and TAK-228 (67%, *p*-value < 0.01). In addition, a significant decrease in proliferation was also observed in trastuzumab with TAK-228 1 nM combination therapy (77% vs. 84% for trastuzumab and vs. 102% for TAK-228 1 nM, *p*-value < 0.05). In the SK-BR-3.rT1 line, the combination of trastuzumab and TAK-228 5 nM decreased growth very significantly (44%) compared to treatment with trastuzumab (96%) and TAK-228 (77%, *p*-value < 0.001). The same effect was observed in the AU-565.rT2 line, with reduced proliferation in combination therapy (64% vs. 100% for trastuzumab, and 84% for TAK-228, *p*-value < 0.001). Finally, in EFM-192A.rT1, a significant decrease in proliferation was identified trastuzumab plus TAK-228 5 nM combined therapy compared to individual treatments (65%, *p*-value < 0.01).

To test the effect of BEZ235 in combination with trastuzumab on cell proliferation, three concentrations of the drug (1 nM, 5 nM and 20 nM) were selected, all below the IC50 value for all lines. The effect on cell proliferation was assessed in the four trastuzumab-sensitive and trastuzumab-acquired resistance lines (Figure 3). Using a BEZ235 concentration of 20 nM, a significant decrease in proliferation was observed in BT-474.rT3 (19%, *p*-value < 0.001) and EFM-192A.rT1 (30%, *p*-value < 0.001) compared to control and trastuzumab treatment conditions. Furthermore, in BT-474.rT3, the combined treatment of BEZ235 plus trastuzumab significantly reversed trastuzumab resistance compared to the trastuzumab treatment condition (45%, *p*-value < 0.001). In sensitive cell lines, trastuzumab combined with BEZ235 20 nM potentiated the effect of trastuzumab individually, with no significant effect.

Two concentrations of everolimus (0.5 nM and 1 nM) were selected below the IC50 value in all cell lines (Figure 4). Treatment with either concentration of the drug alone had no effect on cell proliferation in any of the sensitive or acquired-resistant lines. Combination therapy of trastuzumab with 0.5 nM everolimus showed only slightly stronger effects than trastuzumab alone on proliferation in most cell lines, both sensitive and resistant. However, in the combined condition consisting of trastuzumab and everolimus 1 nM, a reversal of trastuzumab resistance was observed, very significantly decreasing proliferation in the BT-474.rT3 (20%, *p*-value = 0.003) and EFM-192A.rT1 (42%, *p*-value = 0.005) lines, compared to trastuzumab-alone treatment. Furthermore, this treatment combination enhanced the effect of trastuzumab in the four sensitive lines (i.e., BT-474 (13%), SK-BR-3 (26%), AU-565 (35%) and EFM-192 A (31%)), decreasing their proliferation compared to the single-treatment conditions, without being statistically significant (Figure 4).

### 3.4. Potentiation Effect between Trastuzumab and mTORC1/2 Inhibitor TAK-228 in Breast Cancer Cell Lines with Primary Trastuzumab Resistance

The effects of drug combinations on BCCLs with primary resistance to trastuzumab were markedly dependent on each particular cell line (but less so on the nature of the inhibitor, Appendix A). In HCC1419, the combination of trastuzumab with any of the inhibitors had a greater effect than treatment with the inhibitor alone but was generally not effective with respect to treatment with trastuzumab, possibly because at baseline these cells are somewhat sensitive to trastuzumab. In the case of HCC1954, a significant effect was observed in the combination of trastuzumab with any inhibitor, both with respect to trastuzumab and the inhibitor alone. However, JIMT-1 cells showed minimal response to the different treatments, except for a small decrease in cell proliferation, due to the effect of the combination of trastuzumab with TAK-228.

Treatment with BEZ235 at any of the three concentrations tested in combination with trastuzumab resulted in a significant decrease in proliferation in the HCC1954 line. A 65% decrease in proliferation was observed in the BEZ235 1 nM plus trastuzumab condition compared to the single-treatment conditions (*p*-value < 0.001). In the BEZ235 5 nM plus trastuzumab combination, 16% proliferation was identified compared to BEZ235 5 nM treatment (29%) and trastuzumab treatment (87%), with a significant reduction in proliferation (*p*-value < 0.01). Finally, 8% proliferation was observed in the BEZ235 20 nM plus trastuzumab combination, compared to the single treatments (*p*-value < 0.01). JIMT1 cell proliferation was not affected by any of the treatment conditions.

For everolimus, two concentrations (0.5 nM and 1 nM) were selected below the IC50 value in the cell lines, except in JIMT-1. Its effect on cell proliferation was evaluated in the untreated condition, treatment with trastuzumab 15 µg/mL, everolimus 0.5 nM or 1 nM and the combination of both treatments at the two selected everolimus concentrations (Appendix A). Treatment with everolimus at 0.5 nM demonstrated a significant effect on cell proliferation in the HCC1954 line, in combined treatment with trastuzumab (59%, compared to 84%, *p*-value < 0.01), reversing trastuzumab resistance. In addition, treatment with everolimus 1 nM significantly reduced the proliferation of this line (12%, *p*-value < 0.001). In the JIMT-1 cell line, treatment with everolimus 0.5 nM, both alone and in combination with trastuzumab, showed no effect on cell proliferation, while treatment with everolimus 1 nM resulted in a significant reduction in cell proliferation (44%, *p*-value < 0.01).

The treatment effect of the TAK-228 inhibitor was evaluated using two treatment concentrations, 1 nM and 5 nM, in monotherapy and in combination with trastuzumab. This resistance reversal effect was also observed in the primary resistant line HCC1954. Combination therapy with trastuzumab and 5 nM TAK-228 significantly reduced cell proliferation compared to trastuzumab (57% vs. 85%, *p*-value < 0.001) and TAK-228 (57% vs. 81%, *p*-value < 0.001). Cell proliferation of the JIMT-1 line was not modified by any of the treatment conditions tested.

### 3.5. Downregulation of PI3K/AKT/mTOR and MAPK Signalling by the Combination of Trastuzumab with TAK-228 in HER2-Positive Breast Cancer Cell Lines

Since treatment with the inhibitor TAK-228 was shown to reverse trastuzumab resistance in the four cell lines with acquired resistance in combination with trastuzumab, the effect of the combination of both treatments on inhibition of the PI3K/AKT/mTOR pathway was evaluated. The molecular effect of the treatment was assessed by analysing the phosphorylation of the effector proteins of the two mTOR complexes: p-S6 (Ser235/236), p-4E-BP1 (Thr37/46) and p-4E-BP1 (Thr70) of the mTORC1 complex; and p-AKT (Ser473) of the mTORC2 complex, as well as their total forms; in addition, the analysis of the phosphorylated form of ERK was included. Protein expression profiling was performed after 24 h of treatment with trastuzumab 15 μg/mL, or treatment with TAK-228 5 nM, with TAK-228 50 nM, or the combination of trastuzumab plus TAK-228 at the two concentrations above, as well as the control condition.

Combination treatment of trastuzumab with TAK-228 (at either of the two concentrations tested) resulted in a decrease in AKT phosphorylation levels (Ser473) in the BT-474 line, but not in the BT-474.rT3 line (Figure 5A). In both lines, combined treatment with TAK-228 5 nM plus trastuzumab resulted in a significant reduction in p-S6 (Ser235/236) compared to the monotherapy condition, although this reduction was not observed in the two phosphorylated forms of 4E-BP1. In the 50 nM TAK-228 treatment condition, combination with trastuzumab induced disappearance of p-S6 (Ser235/236) and a significant reduction of p-4E-BP1 (Thr37/46 and Thr70) levels in both sensitive and resistant cells. In addition, only in the sensitive line did we observe that TAK-228 combined with trastuzumab resulted in a decrease in the phosphorylated form p-ERK1/2 (Thr202/Tyr204) compared to the levels detected in the single-treatment conditions. Furthermore, the combination of trastuzumab plus TAK-228 5 nM in the sensitive cell line induced a decrease in HER2 phosphorylation levels, while in the BT-474.rT3 line, it was necessary to increase the concentration of the inhibitor to 50 nM (in combination with trastuzumab) to observe the same effect in reduced p-HER2 levels. In both lines, TAK-228 5 nM increased p-HER3, as previously described, and combined treatment with both concentrations of TAK-228 reduced phosphorylation only in the resistant line.

In SK-BR-3 and SK-BR-3.rT1 lines, combined treatment consisting of TAK-228 50 nM and trastuzumab reduced p-AKT levels (Ser473) compared to baseline and trastuzumab treatment, with no change in total form expression (Figure 5B). In both lines, treatment with TAK-228 plus trastuzumab was also found to decrease S6 (Ser235/236) phosphorylation compared to levels detected in the treatment conditions alone. In addition, the 50 nM TAK-228 treatment condition and the trastuzumab combination condition resulted in a highly significant decrease in S6 (Ser235/236) activation, as did the phosphorylated forms of 4E-BP1 (Thr37/46 and Thr70). It is also noteworthy that the total forms of S6 and 4E-BP1 were affected by treatment with TAK-228 50 nM and the combination with trastuzumab. Treatment of both sensitive and resistant cells with TAK-228 alone or in combination with trastuzumab induced an increment in HER2 and HER3 phosphorylation.

In AU-565 and AU-565.rT2 lines, treatment with TAK-228 in combination with trastuzumab resulted in decreased phosphorylation of AKT (Ser473) and S6 (Ser235/236) (Figure 5C). In addition, single TAK-228 treatment lowered the level of p-S6 (Ser235/236) compared to baseline. As in the sensitive and resistant SK-BR-3 lines, the total form of 4E-BP1 decreased in the presence of TAK-228 treatment at either of the two concentrations tested and in combination with trastuzumab, as did the phosphorylated form of 4E-BP1 (Thr37/46). In these lines, the phosphorylation levels of 4E-BP1 (Thr70) are almost undetectable, and no differences between treatment conditions were in evidence. We observed an increase in p-HER2 levels in AU-565 cells when treated with TAK-228 at 5 or 50 nM in combination with trastuzumab. However, TAK-228 50 nM plus trastuzumab in the resistant cell line induced a reduction in phosphorylation levels. Regarding the levels of HER3 phosphorylation, we did not observe a decrease with the different combinatorial treatments in either cell line.

Similarly, in the EFM-192A and EFM-192A.rT1 lines, treatment with TAK-228 at 5 nM and 50 nM and combination with trastuzumab resulted in inhibition of the PI3K/AKT/mTOR pathway (Figure 5D). In both lines, p-AKT (Ser473) levels were found to decrease in the presence of trastuzumab with TAK-228 (at both concentrations) compared to single treatments. In the EFM-192A line, a decrease in p-AKT (Ser473) levels was also observed in the presence of TAK-228 50 nM. In both lines, the combined treatment with TAK-228 50 nM caused a disappearance of p-S6, as well as a decrease in total protein levels. Finally, the EFM19-2A.rT1 line under baseline conditions showed significant activation of p-4E-BP1 (Thr70) compared to its parental line, with very similar levels of total 4E-BP1. Combination treatment with trastuzumab plus TAK-228 50 nM resulted in inhibition of this p-4E-BP1 (Thr70) activation to levels below those of trastuzumab or TAK-228 monotherapy. In addition, as observed in the other cell lines, the levels of the total 4E-BP1 form decreased in the presence of TAK-228 compared to baseline. In the EFM-192A line, as in BT474, combined treatment of trastuzumab with TAK-228 at both concentrations resulted in decreased levels of ERK1/2 (Thr202/Tyr204) phosphorylation compared to levels observed in the single-treatment conditions. In the EFM-192A and EFM-192A.rT1 cells, single or combined treatments did not induce significant changes in HER2 phosphorylation levels. Additionally, we observed an increase in HER3 phosphorylation with the single TAK-228 treatment, though the addition of trastuzumab did not produce a decrease in those levels.

The molecular effect of TAK-228 on the two lines with primary resistance to trastuzumab (i.e., HCC1954 and JIMT-1) was also studied under the treatment conditions mentioned above. In the presence of combined treatment at both concentrations, the HCC1954 line showed a slight decrease in AKT phosphorylation (Ser473) (Appendix A). It was also observed that p-S6 (Ser235/236) was significantly decreased by treatment with TAK-228 at both concentrations, independent of trastuzumab. The same was true for the full form of 4E-BP1 and its phosphorylated form, p-4EBP1 (Thr37/46). In this line, phosphorylation levels of p-4EBP1 (Thr70) were almost undetectable, so no differences between treatments could be assessed. In the JIMT-1 line, only treatment with TAK-228 at either concentration resulted in a trastuzumab-independent decrease in p-S6 (Ser235/236). No changes were observed in 4E-BP1 or its phosphorylated forms, nor in AKT and its phosphorylated form. The original WB images can be found as Supplementary Material (Appendix A).

In summary, the combined treatment decreased the phosphorylation levels of HER2/HER3, diminished PI3K/AKT/mTOR signalling and limited ERK phosphorylation, as a direct consequence of the TAK-228 mechanism of action.

### 3.6. Cell-Cycle and Apoptosis Analysis in Trastuzumab-Resistant Breast Cancer Cell Lines Treated with Trastuzumab and TAK-228

The results of resistance reversal obtained in cell proliferation assays with the combination of trastuzumab plus TAK-228 led us to investigate whether the treatment would also have an impact on cell cycle control, as well as on apoptosis induction. We firstly checked cell viability at shorter times, after treatment with trastuzumab in combination with different concentrations of TAK-228, to discard a deleterious effect. Cell cycle arrest was analysed after treatment with trastuzumab, TAK-228 and the combination of both for 24 h, in the cell lines SK-BR-3, AU-565 and EFM-192A, as well as in their corresponding resistant lines, SK-BR-3.rT1, AU-565.rT2 and EFM-192A.rT1 (Figure 6A). We observed a significant increase in the G0/G1 phase signal in SK-BR-3 and SK-BR-3.rT1 cells treated with the mTOR inhibitor alone (*p* = 0.004, *p* = 0.009, respectively), and the combination with trastuzumab improved the cell cycle delay (*p* = 0.004, *p* = 0.006, respectively). AU-565 and EFM-192A lines showed an increase in G0/G1 arrest with trastuzumab (*p* = 0.04, *p* = 0.007, respectively) and TAK-228 (*p* = 0.006, *p* = 0.005, respectively) treatment alone, but the effect was enhanced with the combined treatment (*p* = 0.004, *p* = 0.001, respectively). However, in the corresponding resistant lines AU-565.rT2 and EFM-192A.rT1, only TAK-228 (*p* = 0.001, *p* = 0.03, respectively) and both treatments (*p* = 0.0005, *p* = 0.01, respectively) were able to significantly induce cell cycle arrest. No significant changes were detected in the cell lines BT-474 and BT-474.rT3.

Apoptosis was determined by positive staining with annexin V by flow cytometry, including both early and late apoptosis. We analysed the apoptotic effect of each treatment as a single agent and in combination in the BT-474, BT-474.rT3, SK-BR-3 and SK-BR-3.rT1 cell lines (Figure 6B). In BT474 and BT-474.rT3 we observed a significant increase in cell death with TAK-228 alone (*p* = 0.02, *p* = 0.0001, respectively), but the combination of both drugs (*p* = 0.012, *p* = 0.0001) showed a greater rise in cell death. Furthermore, treatment of BT-474.rT3 cells with trastuzumab alone induced a significant increase in the percentage of apoptotic cells (*p* = 0.03). In SK-BR-3 and SK-BR-3.rT1 cell lines trastuzumab did not significantly affect the percentage of apoptotic cells, though the treatment with TAK-228 (*p* = 0.013, *p* = 0.021) or the combination of the two led to a significant increase in cell death.

## 4. Discussion

The development of anti-HER2 targeted therapies to treat patients with HER2-positive breast cancer has proved to be effective in survival in both early and advanced settings. For this reason, trastuzumab has been the standard treatment for HER2-positive breast cancer for more than two decades. Despite this advance, almost all patients eventually experience disease progression on trastuzumab-based therapy, due to de novo or acquired resistance. Aside from alterations in the receptor itself, one mechanism that trastuzumab interferes with HER2 signalling is inhibition of the PI3K/AKT/mTOR signalling pathway [36]. As a logical consequence, among the many causes that have been associated with resistance to anti-HER2 therapies in breast cancer, dysregulations in the signalling of the PI3K/AKT/mTOR pathway seem to play an important role [17,21], as we confirmed in our cellular models of acquired resistance (Figure 1 and Appendix A). As we can see in Appendix A and as previously reported by our group [24], the acquisition of resistance to trastuzumab in these four HER2-positive breast cancer cell lines was associated with an increase in the amounts of p-ERK, p-AKT and p-S6, suggesting a higher level of activation of their PI3K and MAPK pathways and a plausible association with mechanisms of resistance generation in these cell line models. This finding is consistent with previous reports of a correlation between increased activation of the PI3K/AKT pathway and resistance to trastuzumab [31]. Mechanistically, PI3K activation, followed by AKT activation, triggers the release of mTOR from the mTORC1 complex, which in turn activates the S61 and 4E-BP1 proteins. In addition, the complex itself has a negative feedback mechanism, which inactivates AKT [37]. The mTOR protein also localises to the mTORC2 complex, exhibiting direct AKT-activation capability at the Ser473 residue, leading to AKT and BAD activation [38]. Unlike the mTORC1 complex, the activation of this complex appears to be AKT-independent and controlled by RAS/MAPKs [37,38]. At the same time, it has been previously described that PI3K/AKT/mTOR pathway inhibition may result in the activation of compensatory pathways that could reduce the antiproliferative activity of these inhibitors [23,39,40,41]. From a clinical point of view, due to the involvement of this pathway in both HER2-mediated signalling and in the emergence of resistance to HER2-targeted therapies, such as trastuzumab, it would therefore be very interesting to consider inhibiting or modulating this pathway. Because inhibition of the PI3K/AKT/mTOR axis results in enhanced HER2 signalling in HER2-overexpressing breast cancer, especially in increased expression of HER2 and HER3 [23], targeting both pathways could prevent the development of resistance.

However, given the importance of this network in the cellular processes of proliferation, differentiation and apoptosis, its inhibition can be expected to be compensated by hyperactivation of alternative molecular pathways, which would offer the tumour cells escape routes to continue oncogenesis and would eventually lead to the therapy failure. Therefore, it seems logical to test different inhibitors of the pathway, from PI3K to AKT to mTOR (both the mTORC1 and mTORC2 complexes) together with trastuzumab, to see which combination is most effective in controlling tumorigenesis and preventing the development of resistance. We decided to test three different inhibitors covering a broad spectrum of effectors in the pathway, from PI3K to the two mTOR complexes, to ensure the effective blockade of the pathway. One strategy has focused on inhibiting the HER2 signalling pathway more effectively with dual blockade approach. The combined use of trastuzumab and mTOR inhibitors has been shown to be more effective to treat HER2-positive breast cancer than single agents [27]. In addition, receptor tyrosine kinase-dependent ERK1 and ERK2 activation following PI3K/AKT/mTOR inhibition have also been described in preclinical models of HER2-positive breast tumours [23,42]. In these cases, the combination of PI3K/AKT/mTOR inhibitors with an anti-HER2 drug or a MEK inhibitor was more effective than single treatments.

The availability of four cellular models of acquired resistance to trastuzumab over an extended period of time (as well as models of primary resistance), in which we had observed hyperactivation of PI3K/AKT/mTOR pathway markers, led us to explore whether combined suppression of HER2 and PI3K/AKT/mTOR signalling was necessary to achieve optimal therapeutic efficacy, given that there are few such studies in the literature. BEZ235 is an inhibitor of the PI3K/AKT/mTOR pathway with a dual inhibitory capacity of PI3K and mTOR due to the high similarity of the tyrosine kinase domains of both proteins. The combination of trastuzumab plus BEZ235 targets those cells with alterations in the PI3K/AKT/mTOR signalling pathway, due to loss of PTEN or activating mutations in PI3K, while maintaining therapeutic pressure on other cells in the same heterogeneous population that are still sensitive to HER2-targeted drugs [43]. Our results confirm that the addition of BEZ235 overcame resistance to the trastuzumab-only regimen in the sensitive cell lines, some acquired-resistant cells, and in some cells with primary resistance (Figure 3), probably due to its inactivation effect on AKT, S6 and 4E-BP1 phosphorylation [25]. Similarly, several in vivo and in vitro models have shown the efficacy of this combination in restoring sensitivity to HER2-targeted therapy [44,45]. Our results demonstrate that the combination with trastuzumab and BEZ235 significantly results in the reversal of trastuzumab resistance in the primary resistant line HCC1954 (Appendix A). This cell line has an activating H1047R mutation in PI3K, which likely makes it significantly susceptible to BEZ235 treatment [46], and consequently, the combination of BEZ235 with trastuzumab can reverse trastuzumab resistance. However, this did not occur in the JIMT-1 line. This line not only showed loss of PTEN, but also overexpression of mucin 4, which has been described as a mechanism of trastuzumab resistance in breast cancer [47]. The limited effect of trastuzumab plus BEZ235 combination therapy in reversing trastuzumab resistance in the acquired resistance cell lines may be because this dual inhibitor only blocks the action of the mTORC1 complex and not the mTORC2 complex. This results in activation of AKT (Ser473) by the mTORC2 complex and overactivation of the pathway, which may not be affected by PI3K inhibition [39]. In addition, dual PI3K/mTOR inhibition by this drug has been reported to produce compensatory ERK activation, due to activation of receptor tyrosine kinases, such as IGF-1R [23,41]. Despite encouraging results in in vitro and preclinical animal models, few clinical trials with BEZ235 in combination with trastuzumab have been conducted, mainly due to the toxicity of the inhibitor, which causes frequent adverse effects in patients, and high variability in responses to the high doses at which treatment is required.

Everolimus is a rapamycin derivative with mTORC1 complex inhibitory capacity, approved by the FDA to treat postmenopausal patients with ER-positive and HER2-negative metastatic breast cancer. Early phase I trials demonstrated that this drug, in combination with trastuzumab, resulted in decreased cell proliferation in trastuzumab-sensitive cell lines [48]. These results were not confirmed in patient cohorts, such as the phase III BOLERO-1 trial [49], but were confirmed in other trials, such as BOLERO-3 [50]. Given that the patient safety profile of everolimus is superior to that of BEZ235, our results at the cellular level are of interest, although its antiproliferative effects were not as pronounced (Figure 4). This difference between everolimus and BEZ235 in terms of cell growth reflects the different mechanisms of action of the drugs in cell lines with different mutational profiles, as reported previously [25]. Over a decade ago, it was proven that the combination of trastuzumab with everolimus can rescue cancer cells from trastuzumab resistance caused by alterations in the PI3K/AKT/mTOR signalling pathway, with greater efficacy than either agent alone [28]. This is achieved by blocking 4E-BP1 and S6 activation, as well as suppressing AKT activation (which everolimus itself phosphorylates and activates in a feedback loop). Our results showed that the combined treatment of trastuzumab and everolimus in trastuzumab-sensitive lines potentiates, although not significantly, the inhibitory effect of trastuzumab on cell proliferation. However, the combination showed no effect in lines with acquired trastuzumab resistance. Notably, our results demonstrate that in the primary resistant line HCC1954, both combination therapy and individual treatment with everolimus had an impact on cell viability, statistically significantly reversing primary trastuzumab resistance (Appendix A). This may be because the HCC1954 line has the PI3K activating mutation H1047R [31]. But this reversal did not occur in the JIMT-1 line, which has loss of PTEN. In other preclinical models of trastuzumab resistance, trastuzumab and everolimus (or rapamycin) combined therapy obtained better results than either agent alone [27]. Today, combining everolimus with anti-HER2 drugs to decrease tumour activity in HER-2-overexpressing patients with resistance to trastuzumab-based therapy for metastatic breast cancer has proven to be a useful clinical strategy, which has been confirmed in numerous clinical trials [48,51,52]. The limited effect of everolimus observed in our results could be because this dual inhibitor, like BEZ235, is only capable of inhibiting the mTORC1 complex. In addition, inhibition of mTORC1 causes a reactivation loop in the PI3K/AKT/mTOR signalling cascade, due to inhibition of S6, which negatively regulates PI3K activation [53].

Our most conclusive results in cellular models, however, were obtained with the combination of trastuzumab plus TAK-228. TAK-228 is a competitive inhibitor of the ATP domain of mTOR that can simultaneously block the activity of the mTORC1 and mTORC2 complexes. In the three primary trastuzumab-resistant lines and the four lines with acquired resistance, dual blockade of the HER2 and PI3K pathways significantly increased the therapeutic response. In sensitive lines, the association of TAK-228 with trastuzumab significantly decreased cell proliferation and demonstrated, at the molecular level, an ability to block both mTOR complexes, decreasing phosphorylation of all the effectors analysed. Therefore, TAK-228 potentiates the inhibitory effect of trastuzumab on the PI3K/AKT/mTOR pathway (Figure 2). Our results demonstrate that treatment with trastuzumab in combination with TAK-228 results in a statistically significant decrease in cell proliferation in all lines with acquired resistance, and reverses resistance to trastuzumab. Furthermore, at the molecular level, trastuzumab plus TAK-228 combination treatment proves superior to individual treatments, decreasing the activation of PI3K/AKT/mTOR pathway effectors that trastuzumab alone was unable to inhibit. In the primary trastuzumab-resistant cell line, HCC1954, treatment with trastuzumab plus TAK-228 also significantly reversed trastuzumab resistance (Appendix A). The effect at the molecular level shows that TAK-228 can block mTORC1, decreasing phosphorylation of S6 and 4E-BP1, but not the mTORC2 complex, because it does not decrease AKT (Ser473) activation. This effect is different from that reported in the literature for TAK-228 treatment in combination with lapatinib, which causes complete inhibition of S6, 4E-BP1 and AKT (Ser473) phosphorylation in the HCC1954 line [30]. The JIMT-1 line, however, is not affected by any TAK-228 plus trastuzumab treatment condition, which supports the data presented above indicating that this line, in addition to the loss of PTEN, could present mutations in MUC4 that stabilise the HER2/HER3 heterodimer, thus making inhibition with this type of drug useless for reversing resistance [47]. Furthermore, no molecular modification of its phosphorylation pattern was observed with treatment, suggesting that this cell line exhibits a PI3K/AKT/mTOR-independent mechanism of resistance to trastuzumab that results in activation of the pathway even in the presence of specific inhibitors. TAK-228 has shown efficacy in preclinical models of resistant breast cancer when combined with different anti-HER2 therapies [29,30]. In a preclinical model with HER2-positive breast cancer patient-derived xenografts, TAK-228 sensitised tumours to trastuzumab, so that the combination of both drugs strongly suppressed tumour growth [54]. Given that this and other preclinical trials have shown that combination treatment of the dual mTOR inhibitor TAK-228 with trastuzumab is more potent in treating HER2-positive breast cancers than either agent alone, it is hoped that in the coming years, we will see clinical trials that comprehensively translate the biology of these cancers and subsequently explore targeted therapy strategies. Clinical trials combining TAK-228 with other drugs (such as letrozole, alisertib or paclitaxel) are still under way in solid tumours, including breast cancer.

Sensitivity to trastuzumab is related to activating alterations of the PI3K/AKT/mTOR pathway (either by PIK3CA mutations [55], low/loss of expression of PTEN [56] or both). As described above, biochemical analysis of HER2 and PI3K/AKT/mTOR pathway targets confirmed that trastuzumab treatment partially suppressed pathway signalling in sensitive lines, which lack activating alterations in the PI3K/AKT/mTOR pathway (Appendix A). HCC1954 and JIMT-1 cell lines were found to harbour activating alterations of the PI3K pathway, whereas the sensitive cell lines were not (BT-474 presents a nonactivating K111N PIK3CA mutation [55]). In the case of primary resistant lines, which do have activating alterations in the pathway, less phosphorylation is reported to be affected. Treatment with TAK-228 (alone or in combination with trastuzumab) resulted in even greater inhibition of these signals in most cell lines. However, in HER2-positive cell lines with primary resistance to trastuzumab and PI3K mutations, treatment with TAK-228 was shown not to affect cell proliferation, in contrast to treatment with BEZ235. These data suggest that in the presence of PI3K activating point mutations, treatment with BEZ235 in combination with trastuzumab may be superior to combination treatment with TAK-228 plus anti-HER2 therapy [25,44,46]. The mutational status of PI3K and expression of PTEN of the cell lines have been previously described (Cosmic Database) [31].

Here, we demonstrate that dual blockade of HER2 and PI3K/AKT/mTOR signalling is effective in improving the therapeutic response in HER2-positive breast cell lines with long-term induced resistance to trastuzumab. The combination of trastuzumab with TAK-228 significantly increased the therapeutic response in all the cases (Figure 2A), suggesting that a decrease in mTOR activation status by TAK-228, as determined by the reduction in phosphorylation levels of S6 and 4E-BP1 (Figure 5), affects trastuzumab sensitivity. One limitation to our study is that we have considered resistance in single trastuzumab treatment models, when the current therapeutic protocol establishes first-line treatment with trastuzumab in combination with pertuzumab (a second monoclonal antibody) for HER2-positive breast cancer. To address this limitation, we have generated four de novo models of HER2-positive cell lines with acquired resistance to trastuzumab plus pertuzumab combination therapy. It will be interesting to see whether some of these models also exhibit the PI3K/AKT/mTOR pathway hyperactivation characteristic of the models presented here, and if so, whether ablation of this signal by dual treatment with inhibitors, such as TAK-228 (or others) are effective in treating this refractory cancer.

## 5. Conclusions

In summary, our results obtained in models of sensitive breast cancer cell lines, lines with acquired resistance, and lines with primary resistance to trastuzumab, exposed to combination therapy with specific inhibitors of the PI3K/AKT/mTOR signalling pathway plus trastuzumab, suggest that this combination therapy favours the reversal of trastuzumab resistance. Inhibition of the PI3K/AKT/mTOR pathway using the mTORC1 and mTORC2 inhibitor, TAK-228, can reverse acquired resistance to trastuzumab in all models generated and in some primary resistant lines. When combined with trastuzumab, treatment with the inhibitor TAK-228 has been shown to be superior to the other two inhibitors tested, BEZ235 and everolimus, in reversing acquired trastuzumab resistance. However, in the presence of PI3K activating mutations, single and combined treatment with BEZ235 has been shown to be superior to treatment with TAK-228 and everolimus.

## Figures and Tables

**Figure 1 cancers-13-02778-f001:**
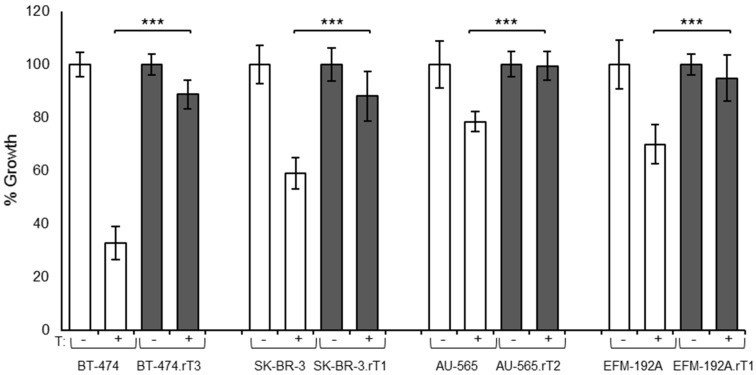
Characterisation of a panel of cell-line models of acquired trastuzumab resistance. Effect of trastuzumab treatment on sensitive and resistant cells. Proliferation was measured after seven days of treatment by trypan blue exclusion. T: Trastuzumab 15 µg/mL. Data are expressed as mean ± SD from ≥ three independent experiments. *** denotes *p* ≤ 0.001.

**Figure 2 cancers-13-02778-f002:**
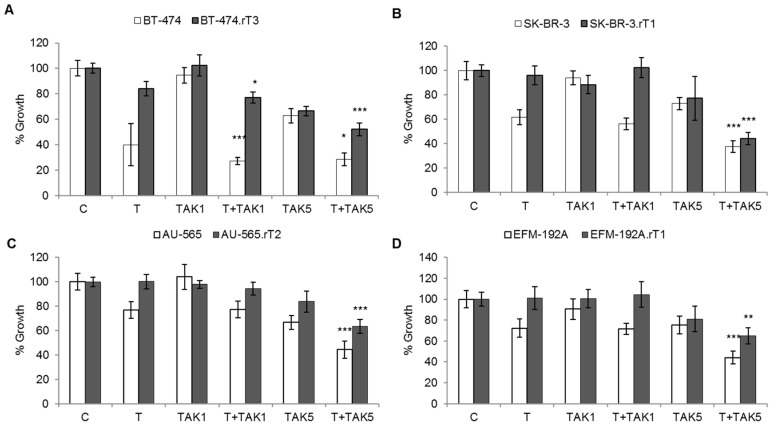
Decrease in mTOR activation status by TAK-228 affects trastuzumab sensitivity. Sensitive and trastuzumab-resistant cells were treated for seven days with DMSO, 15 μg/mL trastuzumab (T), 1 or 5 nM TAK-228 (TAK), or a combination of 15 μg/mL trastuzumab plus 1 or 5 nM TAK-228. Viable cells were then counted by trypan blue exclusion. Viability is presented as a percentage of the DMSO-treated control vector group. Error bars represent standard deviation between replicates (*n* ≥ 3). * denotes *p* ≤ 0.05, ** denotes *p* ≤ 0.01 and *** denotes *p* ≤ 0.001. (**A**) BT-474 sensitive (BT474) and trastuzumab-resistant (BT-474.rT3) cells. (**B**) SK-BR-3 sensitive (SK-BR-3) and trastuzumab-resistant (SK-BR-3.rT1) cells. (**C**) AU-565 sensitive (AU565) and trastuzumab-resistant (AU-565.rT2) cells. (**D**) EFM-192A sensitive (EFM-192A) and trastuzumab-resistant (EFM-192A.rT1) cells.

**Figure 3 cancers-13-02778-f003:**
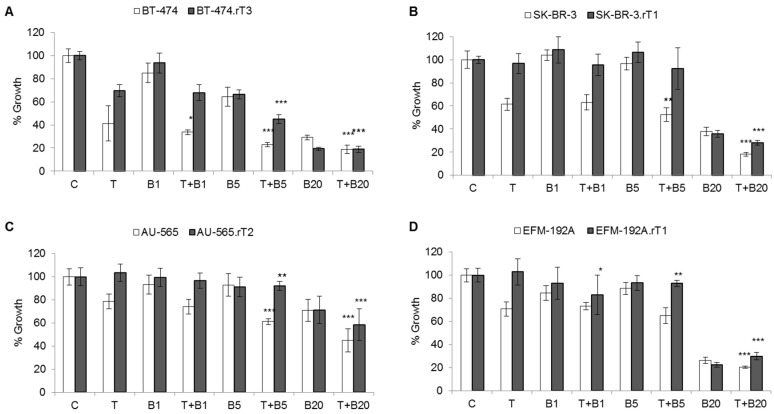
Effect of blocking mTOR activation by BEZ235 on trastuzumab sensitivity in trastuzumab-sensitive and trastuzumab-acquired resistance cell lines. Sensitive and trastuzumab-resistant cells were treated for seven days with DMSO, 15 μg/mL trastuzumab (T), 1, 5 or 20 nM BEZ235 (**B**), or a combination of 15 μg/mL trastuzumab plus 1, 5 or 20 nM BEZ235. Viable cells were then counted by trypan blue exclusion. Viability is presented as a percentage of the DMSO-treated control vector group. Error bars represent standard deviation between replicates (*n* ≥ 2). * denotes *p* ≤ 0.05, ** denotes *p* ≤ 0.01 and *** denotes *p* ≤ 0.001. (**A**) BT-474 sensitive (BT-474) and trastuzumab-resistant (BT-474.rT3) cells. (**B**) SK-BR-3 sensitive (SK-BR-3) and trastuzumab-resistant (SK-BR-3.rT1) cells. (**C**) AU-565 sensitive (AU-565) and trastuzumab-resistant (AU-565.rT2) cells. (**D**) EFM-192A sensitive (EFM-192A) and trastuzumab-resistant (EFM-192A.rT1) cells.

**Figure 4 cancers-13-02778-f004:**
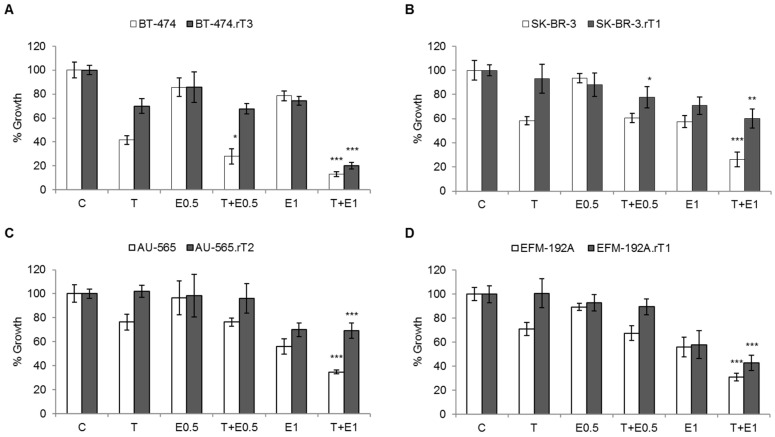
Effect of blocking mTOR activation by everolimus on trastuzumab sensitivity in trastuzumab-sensitive and trastuzumab-acquired resistance cell lines. Sensitive and trastuzumab-resistant cells were treated for seven days with DMSO, 15 μg/mL trastuzumab (T), 0.5 or 1 nM everolimus (E), or a combination of 15 μg/mL trastuzumab plus 0.5 or 1 nM everolimus. Viable cells were then counted by trypan blue exclusion. Viability is presented as a percentage of the DMSO-treated control vector group. Error bars represent standard deviation between replicates (*n* ≥ 2). * denotes *p* ≤ 0.05, ** denotes *p* ≤ 0.01 and *** denotes *p* ≤ 0.001. (**A**) BT-474 sensitive (BT474) and trastuzumab-resistant (BT-474.rT3) cells. (**B**) SK-BR-3 sensitive (SK-BR3) and trastuzumab-resistant (SK-BR-3.rT1) cells. (**C**) AU-565 sensitive (AU-565) and trastuzumab-resistant (AU-565.rT2) cells. (**D**) EFM-192A sensitive (EFM192A) and trastuzumab-resistant (EFM-192A.rT1) cells.

**Figure 5 cancers-13-02778-f005:**
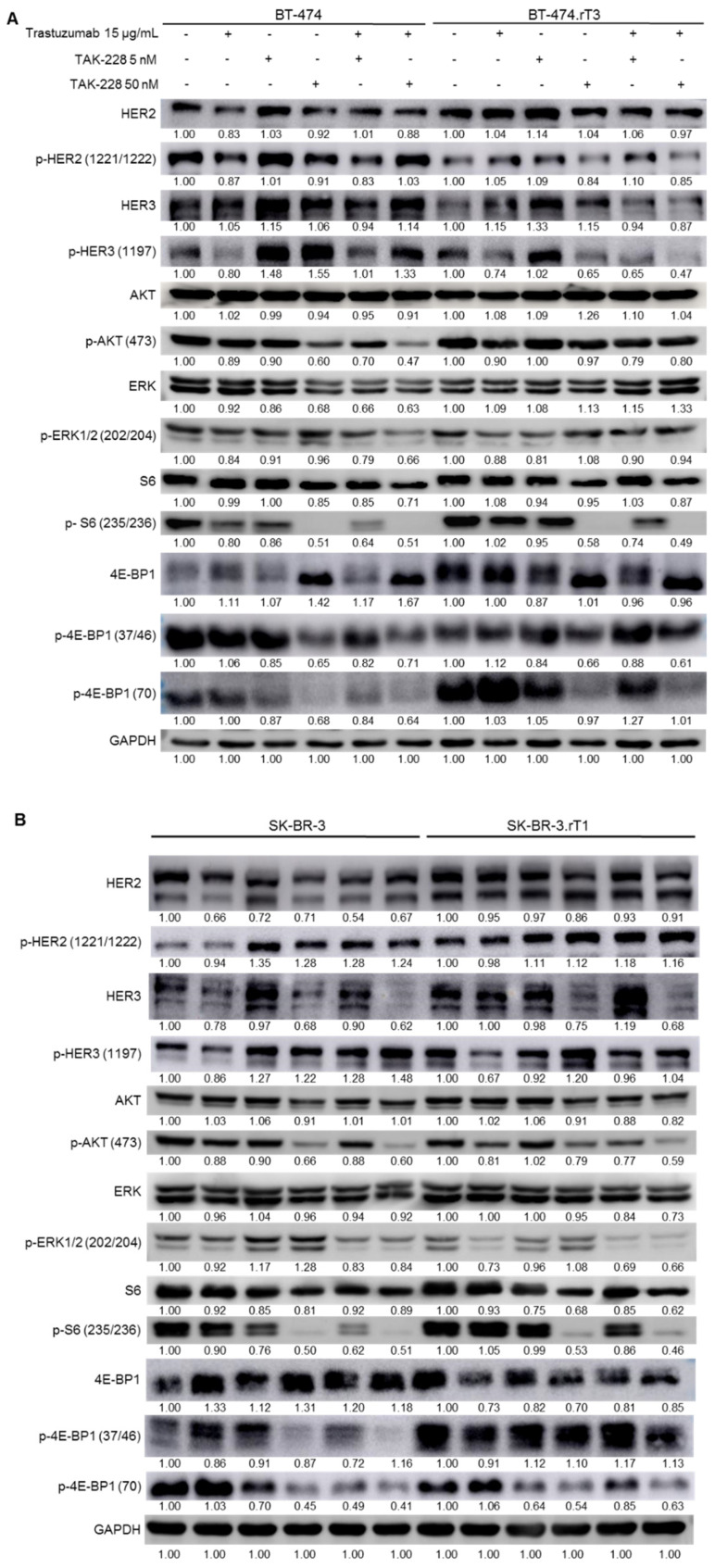
(**A**) Inhibition of p-S6 (Ser235/236) in trastuzumab-sensitive and -resistant BT-474 cells treated with a combination of trastuzumab and TAK-228. Sensitive and trastuzumab-resistant cells were treated for 24 h with DMSO, 15 μg/mL trastuzumab (T), 5 and 50 nM TAK-228 (TAK), or a combination of 15 μg/mL trastuzumab plus 5 or 50 nM TAK-228. Whole-cell protein extracts were analysed with the indicated antibodies. Images are representative of three independent experiments. (**B**) Inhibition of p-S6 (Ser235/236) in trastuzumab-sensitive and -resistant SK-BR-3 cells treated with a combination of trastuzumab and TAK-228. Sensitive and trastuzumab-resistant cells were treated for 24 h with DMSO, 15 μg/mL trastuzumab (T), 5 and 50 nM TAK-228 (TAK), or a combination of 15 μg/mL trastuzumab plus 5 or 50 nM TAK-228. Whole-cell protein extracts were analysed with the indicated antibodies. Images are representative of three independent experiments. (**C**) Inhibition of p-S6 (Ser235/236) in trastuzumab-sensitive and -resistant AU-565 cells treated with a combination of trastuzumab and TAK-228. Sensitive and trastuzumab-resistant cells were treated for 24 h with DMSO, 15 μg/mL trastuzumab (T), 5 and 50 nM TAK-228 (TAK), or a combination of 15 μg/mL trastuzumab plus 5 or 50 nM TAK-228. Whole-cell protein extracts were analysed with the indicated antibodies. Images are representative of three independent experiments. (**D**) Inhibition of p-S6 (Ser235/236) in trastuzumab-sensitive and -resistant EFM-192A cells treated with a combination of trastuzumab and TAK-228. Sensitive and trastuzumab-resistant cells were treated for 24 h with DMSO, 15 μg/mL trastuzumab (T), 5 and 50 nM TAK-228 (TAK), or a combination of 15 μg/mL trastuzumab plus 5 or 50 nM TAK-228. Whole-cell protein extracts were analysed with the indicated antibodies. Images are representative of three independent experiments.

**Figure 6 cancers-13-02778-f006:**
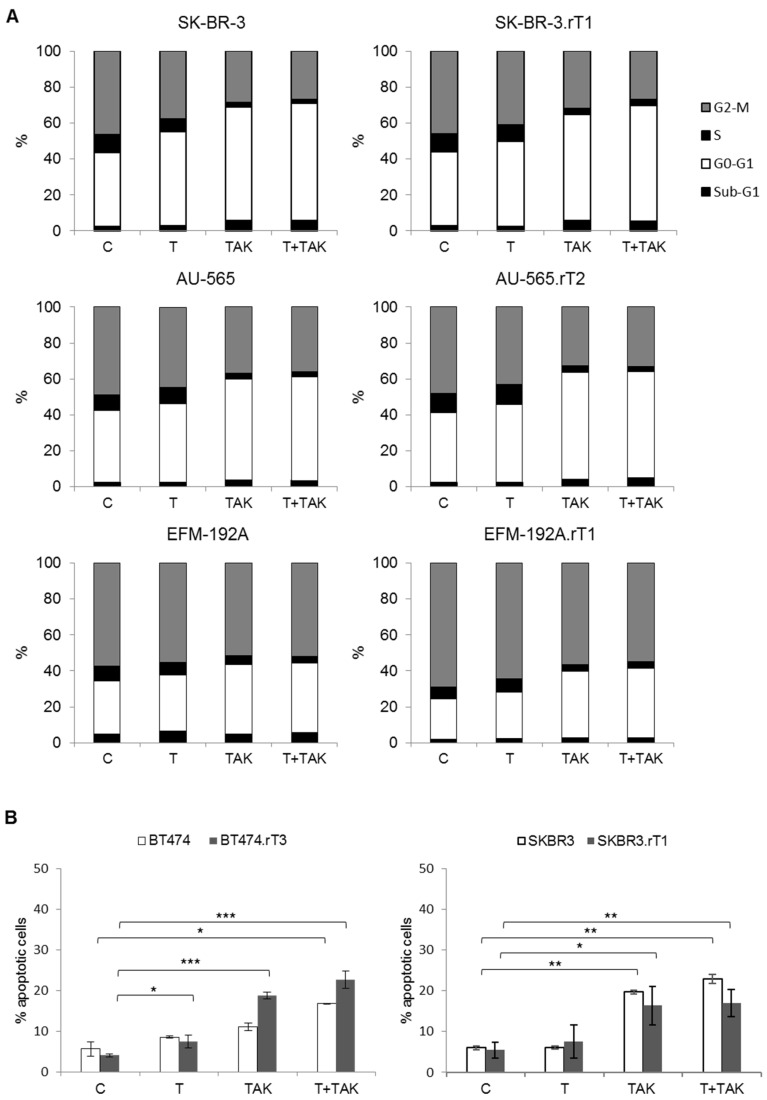
(**A**) Cell cycle arrest induced by trastuzumab and TAK-228 in trastuzumab-sensitive and -resistant cell lines. Cell lines were treated with 15 µg/mL trastuzumab (T), 0.5 µM TAK-228 (TAK) or a combination (T+TAK). Cell cycle arrest was analysed by flow cytometry after 24 h. (**B**) Apoptosis induced by trastuzumab and TAK-228 in trastuzumab sensitive and resistant cell lines. Cell lines were treated with 15 µg/mL trastuzumab (T), 0.5 µM TAK-228 (TAK) or the combination (T+TAK). Apoptosis was measured after 72 h by Annexin V positive staining by flow cytometry. Data are expressed as mean ± SD from three independent experiments. * denotes *p* ≤ 0.05, ** denotes *p* ≤ 0.01 and *** denotes *p* ≤ 0.001.

**Table 1 cancers-13-02778-t001:** Inhibitory concentrations of mTOR inhibitors as a measure of proliferation inhibition in a panel of breast cancer cell lines.

Cell Line	Proliferation IC50 (nM)			
	BEZ235	Everolimus	TAK-228	Sensitivity to Trastuzumab
BT-474	3.4	3.7	9.4	S
BT-474.rT3	2.4	3.8	6.1	R
SK-BR-3	6.3	3.2	5.3	S
SK-BR-3.rT1	6.4	5.5	8.9	R
AU-565	18.0	7.5	13.1	S
AU-565.rT2	12.8	1.8	9.3	R
EFM-192A	3.6	5.9	5.9	S
EFM-192A.rT1	2.2	1.8	7.6	R
HCC1419	33.2	0.7	6.7	S/R
HCC1954	27.3	23.2	12.8	R
JIMT-1	17.9	0.1	21.0	R

Note: A panel of HER2-positive breast cancer cell lines was treated with escalating concentrations of BEZ235, everolimus and TAK-228. Proliferation was measured by counting cells after seven days of treatment. Viable cells were counted by trypan blue exclusion. IC50 (half-maximal effective concentration) was calculated using the SigmaPlot software. Values are mean IC50 from three independent experiments. BT-474: BT-474 trastuzumab-sensitive cells. BT-474.rT3: BT-474 trastuzumab-resistant cells. SKBR3: SKBR3 trastuzumab-sensitive cells. SK-BR-3.rT1: SK-BR-3 trastuzumab-resistant cells. AU-565: AU-565 trastuzumab-sensitive cells. AU-565.rT2: AU-565 trastuzumab-resistant cells. EFM-192A: EFM-192A trastuzumab-sensitive cells. EFM-192A.rT1: EFM192A trastuzumab-resistant cells.

## Data Availability

Data sharing is not applicable to this article.

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
