# Peer review of "The Novel Oral mTORC1/2 Inhibitor TAK-228 Reverses Trastuzumab Resistance in HER2-Positive Breast Cancer Models"

_cancers, 2021, doi:10.3390/cancers13112778_

Round 1

Reviewer 1 Report

The manuscript by Rojo and coworkers describes their investigation of the impact of TAK-228 on trastuzumab resistance in HER2-positive breast cancel models. It is well known that breast cancer continue to a deadly risks to the lives of many patients worldwide. The subtype HER2 accounts for about 25% of all cases. Although the drug trastuzumab has demonstrated remarkable efficacy, the main issue is the primary or acquired resistance emerges within only a few months after the start of treatment of breast cancer patients. I think the research presented in this manuscript would be valuable to the readers of Cancers journal. Overall, the paper is well written and the results are well presented. I recommend publishing after some minor revisions.

  1. Page 6, line 270-271, why are these concentrations chosen? It was mentioned that the concentration and time of treatments were based on previous reports. But no references of previous reports were given here.
  2. On page 9, the paragraph starts with line 331. Two concentrations of everolimus 0.5 nM and 1nM were selected and was found that 0.5 nM of this drug has no effect while 1.0 nM caused a significant decrease in growth in BT-474.rT3 when compared to the treated control (26%). But the combination of trastuzumab and everolimus 1nM combination treatment condition, caused a decrease in growth in BT-474.rT3 (20%). From this data, it seems that the combination doesn’t improve the decrease in growth in BT-474.rT3. It would be interesting to see what would be the effect of everolimus at different concentrations when used alone without the trastuzumab. For example, what is the effect of higher concentration of everolimus (> 1 nM)?

Reviewer 2 Report

It is a thorough study.

The study identifies that one of the mechanisms by which HER-2 inhibitor trastuzumab causes resistance in HER-2 positive breast cancer cell lines is by upregulating PI3kianse/Akt/mTOR pathway (specifically pS-6). This is a relevant finding in the context that most of the drugs designed as targetted signaling molecule inhibitors, after prolong use, cause drug resistance. It is a major problem for cancer therapy. The experiemnts designed are starightforward. Where as the initial findings are similar to other findings, this study is interesting as the auhtors have shown that pS6 specifically is up regulated (Fig 5, A-D, more phosphorylation) in almost all resistant cells, which is down stream of Akt pathway, and which is inhibited by the combination. It will be a better idea if the authors change the titles of the description of figures in the results and specifically state where it works (not just mentiong pathway). In addition to pS-6 inhbition, the authors found that mTOR inhibitors sensitized these resiatant breast cancer cells to trastuzumab. Though the inhibition results are not drastic (Fig2, A-D, ) indicating that other pathways/mechanisms may also play role in this process, the results are encouraging. The manuscript cited relevant papers and this manuscript is acceptable as it is expected to generate further research. The authors may improve the manuscript by discussing and comparing their results with other combination therapy studies which are published.
